# Exercise-Induced Arrhythmia or Munchausen Syndrome in a Marathon Runner?

**DOI:** 10.3390/diagnostics13182917

**Published:** 2023-09-12

**Authors:** Robert Gajda, Wojciech Drygas, Jacek Gajda, Pawel Kiper, Beat Knechtle, Magdalena Kwaśniewska, Maciej Sterliński, Elżbieta Katarzyna Biernacka

**Affiliations:** 1Center for Sports Cardiology at the Gajda-Med Medical Center in Pułtusk, ul. Piotra Skargi 23/29, 06-100 Pułtusk, Poland; j.gajda@gajdamed.pl; 2Department of Kinesiology and Health Prevention, Jan Dlugosz University, 42-200 Czestochowa, Poland; 3Faculty of Medicine, Lazarski University, ul. Swieradowska 43, 02-662 Warsaw, Poland; wdrygas@ikard.pl; 4National Institute of Cardiology, ul. Alpejska 42, 04-628 Warszawa, Poland; msterlinski@ikard.pl (M.S.); k.biernacka@ikard.pl (E.K.B.); 5Healthcare Innovation Technology Lab, IRCCS San Camillo Hospital, Via Alberoni 70, 30126 Venice, Italy; pawel.kiper@hsancamillo.it; 6Institute of Primary Care, University of Zurich, 8091 Zurich, Switzerland; beat.knechtle@hispeed.ch; 7Medbase St. Gallen Am Vadianplatz, 9000 St. Gallen, Switzerland; 8Department of Preventive Medicine, Faculty of Health Sciences, Medical University of Lodz, ul. Lucjana Żeligowskiego 7/9, 90-752 Łódź, Poland; magdalena.kwasniewska@umed.lodz.pl

**Keywords:** heart rate monitor, endurance running, arrhythmia, tachycardia, artifacts, exertion rhythm disorders, Munchausen syndrome, athlete’s heart

## Abstract

A 36-year-old professional marathon runner reported sudden irregular palpitations occurring during competitions, with heart rates (HR) up to 230 bpm recorded on a sports HR monitor (HRM) over 4 years. These episodes subsided upon the cessation of exercise. Electrocardiograms, echocardiography, and cardiac magnetic resonance imaging results were borderline for athlete’s heart. Because an electrophysiology study and standard exercise tests provoked no arrhythmia, doctors suspected Munchausen syndrome. Ultimately, an exercise test that simulated the physical effort of a competition provoked tachyarrhythmia consistent with the HRM readings. This case demonstrates the diagnostic difficulties related to exercise-induced arrhythmia and the diagnostic usefulness of sports HRMs.

## 1. Introduction

Though regular exercise training is considered an important health-promoting and disease-preventive measure, strenuous long-lasting exercise may lead to unfavorable changes, including heart rate (HR) disturbances [1,2]. Many cardiologists are of the opinion that extreme endurance exercises performed by professional and even ambitious leisure-time athletes could be dangerous [3,4] and even provoke sudden cardiac death [5,6]. Wearable HR monitors (HRMs) are increasingly used by professional and leisure-time athletes, as well as by cardiac patients, to monitor HR during exercise [7,8,9]. A HRM may notify the user if it detects abnormal HR; thus, it potentially protects against dangerous incidents [10,11]. On the other hand, such devices may also sometimes provide false information, suggesting serious heart rhythm disturbances during exercise even in healthy, asymptomatic individuals [12,13]. Diagnosis is in some cases very difficult, as is illustrated by the present case concerning a professional marathon runner.

## 2. Case Presentation

### 2.1. Subject Characteristics and Health History

A 36-year-old marathon runner (height, 1.73 m; weight, 66 kg; body mass index, 21.72 kg/m^2^) came to the Center for Sports Cardiology (CSC) in Pułtusk, Poland, seeking a diagnosis for his cardiac problems. The athlete has led an active life since childhood, has been running regularly for 20 years, and has run approximately 90,000–100,000 km in his lifetime. As part of his daily training over the past year, he has run an average of 20 km/day. During his 20-year sports career, he has successfully competed many times in Polish, European, and World Championships, very often winning medals in long-distance running, marathons, and ultramarathons (both cross-country, mountain, and street running). An extensive sports biography and the athlete’s personal records can be found on the official website of the Polish Athletics Association and in Appendix A (Appendix A).

Before 2019, the athlete had never experienced any noticeable arrhythmia. He had used HRMs for many years, with no unexpected HR spikes either when training or competing. As a junior in 2004 and 2005, however, he fainted twice while competing. Then, in 2019, during an intense training run in Kyrgyzstan (1700 m above sea level), his HRM recorded a tachyarrhythmic seizure for the first time, with his HR suddenly increasing to over 200 bpm. This tachyarrhythmia made it impossible to continue training but subsided after about 30 s. While in Kyrgyzstan, he suffered severe pneumonia, confirmed radiologically after the end of the sports camp. Afterward, he continued experiencing instances of tachyarrhythmia, almost exclusively during competitions. Each time, the athlete felt a sensation of irregular heartbeat and observed an increase in his HR on his HRM. The arrhythmia generally occurred in the last phase of extreme effort (e.g., at the 30th km of a marathon), when the running pace was around 3:15–3:20 min/km and the HR around 172–175 bpm. A high temperature, rough route, or jerky pace increased the likelihood of arrhythmia. Subsequently, it also happened at lower HR values (Figure 1A,B).

Apart from the documented HRM data, this cardiac arrhythmia was otherwise witnessed only once, by ambulance nurses (but without an electrocardiogram (ECG)) who were called in after the athlete fainted at the 32nd km of the Polish Marathon Championship in 2021 (verbal information). He was afterwards intensively tested in reference cardiology centers, including countless ECGs, two stress tests (on a treadmill and a bicycle), echocardiography, cardiac magnetic resonance imaging (MRI), and, eventually, an electrophysiology study. Neither stress tests nor the electrophysiological study provoked an arrhythmia attack. The test results were considered borderline for an athlete’s heart, thus precluding a diagnosis of heart disease.

Additional attempts were made to record potential arrhythmias in the athlete. The athlete competed in a marathon and a half-marathon twice, simultaneously wearing a Holter ECG and a sports HRM, albeit without showing tachyarrhythmia. Additional exercise tests were performed on a treadmill, without arrhythmias. Furthermore, the readings recorded by the HRM were called into question and treated as artifacts, which are often very difficult to distinguish from an undisturbed recording [14,15] (Figure 1C). The clinical data from the patient’s history, although quite convincing, were not confirmed by the diagnostic tests already mentioned, in excess of the indications from sports HRM. Some doctors suspected the athlete exhibited Munchausen syndrome (MS). Meanwhile, the athlete continued to compete and experienced arrhythmia attacks that effectively disqualified him from competing for top places. Arrhythmias of up to 230 bpm on his sports HRM appeared with increasing frequency, with concomitant clinical complaints. By the end of 2022, instances of arrhythmia were occurring at shorter distances (10 km) and sometimes also during training sessions.

The runner provided written informed consent to participate in the analysis and for his data to be published.

### 2.2. Study Protocol

We performed a treadmill exercise test following an individualized protocol, with the main goal of provoking an arrhythmia. An additional goal was to assess the reliability of the sports HRM the athlete used daily by comparing its measurements to those indicated on the ECG during the exercise test. When the maximum treadmill speed was reached, the treadmill angle was adjusted (between 10 and 18%). The exercise was continued until refusal or provoked arrhythmia. During the test, the athlete wore his normal sports HRM (Smartwatch Suunto 9 Peak with Suunto Smart Heart Rate Belt (Suunto Oy, Vantaa, Finland)) in order to assess the consistency of HRM readings during exercise. We used ECG indications from the exercise testing system as a HR reference. Figure 2 shows the athlete with the tested HRM.

Some of the sports HRM measurements were tested and compared with a Holter ECG. A team of doctors and technicians with extensive experience in the analysis of HRM measurements used by athletes of various disciplines training under different conditions [16,17,18] analyzed the results.

### 2.3. Test Results

In minute 17 of the treadmill exercise stress test, the maximum treadmill speed (9.7 km/h) was reached, and the treadmill angle was adjusted (between 10 and 18%). A stable heart rate of 172–175 bpm was reached in minute 15. The highest temporary metabolic equivalent of task (MET) load of 18.1 was achieved in minute 18 and corresponded to 63 mL/kg/min VO2max. In minute 25, a brief supraventricular tachycardia (SVT) paroxysm of several seconds was noted on both devices simultaneously, with a rhythm of 187 bpm (Figure 3A (arrow 1) and Figure 3B). The exercise was continued by changing the angle of the treadmill so that the athlete continued as long as possible at a submaximal load. In minute 28, the SVT paroxysm occurred again at a rate of 230 bpm, with the same rhythm being indicated by the HRM (Figure 3A (arrow 2) and Figure 3C). The athlete continued the effort for more than one minute without slowing down, despite persistent irregular tachyarrhythmia (possible atrial fibrillation), increasing weakness, and a strong feeling of palpitations. The tachyarrhythmia ceased immediately after the exercise stopped. No typical ischemic ST changes were observed. Multiple increasing supraventricular and single ventricular beats were noted during exercise, as were numerous artifacts resulting from vibrations. Both the HRM and the Holter ECG showed the same HR values.

We had also previously tested the accuracy of the athlete’s HRM measurements when worn simultaneously with a Holter ECG. The HRM showed the same maximum, minimum, average, and momentary values of HR indicated by the Holter ECG.

## 3. Discussion

We confirmed the presence of cardiac arrhythmia in an experienced high-performance marathon runner. Despite having always been considered healthy, unexpected increases in HR readings on his sports HRMs in recent years, combined with a feeling of palpitations and weakness, were preventing him from continuing endurance efforts. Numerous ECGs (Appendix A), Holter ECGs (S2), an echocardiography study (Video S3), an MRI (Figure and Video S4), several treadmill and bicycle exercise tests, and finally a cardiac electrophysiology study (Appendix A) (in Appendix A) failed to reveal cardiac arrhythmias. These tests also provided no grounds for establishing a diagnosis excluding the athlete from further sports competition. MRI showed no significant areas of late gadolinium enhancement (LGE), though small pin spots of LGE could be observed in the basal-to-mid-lateral wall and inferior septum. Discrete focal areas of fibrosis marked by LGE are often found in athletes and their clinical importance is considered to be limited [19].

The diagnostic tests performed showed moderate enlargement of all heart cavities as well as slight hypertrophy of the left ventricular walls. Given that the athlete is a member of the national team in ultramarathon running, the occurrence of features of the “athlete’s heart”, to which the above changes correspond, seem to be normal and even expected [20,21,22]. On the other hand, the above changes can also occur in the early stages of hypertrophic cardiomyopathy and arrhythmogenic cardiomyopathy [23,24,25].

Definitive confirmation of the presence of cardiac arrhythmias provoked by extreme exercise diagnosed solely on the basis of sports HRM is essential to avoid the potential risk of sudden cardiac death during sports or a false diagnosis. Gajda et al. proposed an algorithm for how to proceed in situations involving “rhythm disturbances” provoked by exertion, observed on a sports HRM to distinguish true cardiac arrhythmia from mere artifacts (Figure 4) [26].

Given that no diagnosis could be confirmed in the case of this particular athlete, the possibility of the symptoms being simulated in order to justify his lack of success in competitions (MS) was entertained. MS, also called “factitious disorder imposed on self”, is a psychiatric disorder in which a person assumes the role of a sick patient without the intention of external gain (e.g., time off from work, medications) [27]. A number of cases of MS have been described in the literature, including patients who simulate cardiac symptoms, such as HR disturbances (cardiopathia fantastica) [28,29,30]. Diagnosis is always difficult, and in several cases, patients have visited numerous hospitals and sometimes been subjected to unnecessary, costly, and even risky diagnostic procedures [29]. Clarke and Melnick (in 1958) examined cases of this sort and concluded that “all the patients are psychopaths” [31]. However, after careful consideration, MS was definitively ruled out in the present case.

We decided to use an individualized protocol designed to replicate a prolonged run during a sports competition. During the exercise test, the pace and incline of the treadmill were altered to sustain submaximal effort for as long as possible with the goal of inducing arrhythmia, which did indeed occur. The arrhythmia described was supraventricular tachycardia with a ventricular rhythm of 230 bpm and appeared permanently at a HR of 172–175 bpm in minute 28 of the test (corresponding to kilometer 10 in a real race). The athlete was able to continue at the same load level for one more minute, despite rapidly increasing fatigue and a feeling of palpitations. The tachycardia did not subside spontaneously, but only when the effort ceased. The available ECG recorded during the stress test poses several limitations to determining the mechanism of the arrhythmic episode. A sudden heart rate increase can indeed be observed, with its onset and resolution, its regularity, and a slight change in QRS morphology. These findings do not legitimate a diagnosis of atrial fibrillation, focal atrial tachycardia, or other regular focal/reentry arrhythmias. The depicted findings only show that “something corresponding to the reported events did really occur”.

The enlargement of both atria in the ultramarathon runner we studied, which was recognized by both echocardiography and MRI, although typical of an athlete’s heart, suggests that atrial fibrillation should be considered as a potential exercise-induced arrhythmia. More generally, arrhythmia in the context of biatrial enlargement in athletes is still a subject of discussion and equivocal conclusions [32,33].

According to D’Souza et al., athletes are prone to supraventricular rhythm disturbances—these include sinus bradycardia, heart block, and atrial fibrillation. Mechanistically, this is attributed to high vagal tone and cardiac electrical and structural remodeling. However, a relationship between atrial enlargement and AF has yet to be established in athletes. An association between dilatation and AF prevalence was not observed in young athletes, despite 20% of the cohort presenting left atrial enlargement [34].

Newman et al., in turn, reported that athletes have a significantly greater likelihood of developing AF as compared to non-athlete controls, with those participating in mixed sports and younger athletes at the greatest risk. Future studies of AF prevalence in athletes according to specific exercise dose parameters, including training and competition history, may help clarify these risks [35]. Abdulla et al. similarly found that the risk of AF is significantly higher in athletes as compared to non-athlete controls. However, this finding should be further reaffirmed in large-scale prospective longitudinal studies [36].

Heitmann et al. see left atrial (LA) enlargement as an independent risk factor for atrial fibrillation (AF). They noted that some athletes are at increased risk of AF, which may be linked to LA enlargement; however, little is known about the relationship between LA enlargement and AF risk at a moderate level of physical activity (PA). They concluded that moderate PA was associated with reduced AF risk, and PA attenuated the increased risk of AF with LA enlargement in both men and women and all age groups [37].

Cavigli et al. analyzed the acute exercise-induced effects of ultramarathon racing on atrial remodeling and supraventricular arrhythmias in non-professional master athletes. They reported that, despite the presence of supraventricular arrhythmias, particularly at rest, acute exercise-induced atrial dysfunction was not detected. This suggests that biatrial functional remodeling does not play a role as a substrate for the occurrence of supraventricular arrhythmias in such athletes—contrary to the hypothesis of an acute mechanical biatrial dysfunction induced by ultra-endurance exercise [38].

The above situation accurately reflected what the athlete we studied experienced in running competitions when the attack subsided after stopping or after a significant decrease in running pace but recurred when the athlete tried to resume running at the previous pace. During the test, the athlete wore the same HRM he used to wear in sports competitions, and throughout the entire test, it indicated exactly the same HR values as the ECG from the exercise test. A test performed previously using a Holter ECG also showed a convergence in HR with the HRM, thus eliminating the notion that the HRM readings might be an artifact. Incorrect HRM readings can be so aggravating when participating in long runs that some top athletes quit using HRMs during competition altogether [39,40]. In our opinion, HRM readings should be considered reliable when coincident with clinical symptoms, as in this case. The athlete has been informed that he should absolutely refrain from competing in sports and performing maximal and submaximal efforts until diagnostics are completed and a final diagnosis is established.

## 4. Conclusions

In conclusion, attempting to confirm the presence of arrhythmia induced by extreme endurance exercise in well-trained endurance athletes can in some cases pose a significant diagnostic challenge. Sports HRM indications suggesting paroxysms of rapid tachyarrhythmias during extreme effort in a symptomatic marathon runner cannot be neglected and should be verified during subsequent exercise tests mimicking real-life endurance sports competitions.

## Figures and Tables

**Figure 1 diagnostics-13-02917-f001:**
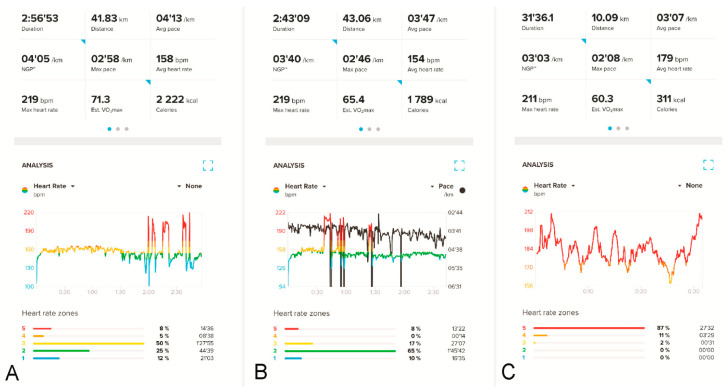
HR recordings on the athlete’s HRM (Suunto Smartwatch 9 Peak with Suunto Smart Heart Rate Belt) during two different marathon runs (**A**,**B**) and a 10 km run (**C**). In (**A**,**B**), HR spikes (red peaks) up to about 219 bpm receded after the athlete stopped for a short period. Panel (**B**) additionally shows the running pace in black; the five sharp “down-spikes” in black indicate momentary stops by the runner during the competition. Panel (**C**) shows apparently similar changes in the runner’s HR values during a 10 km competition (158–210 bpm), but these are simple artifacts; these disturbances resolved after the HRM belt’s battery was replaced.

**Figure 2 diagnostics-13-02917-f002:**
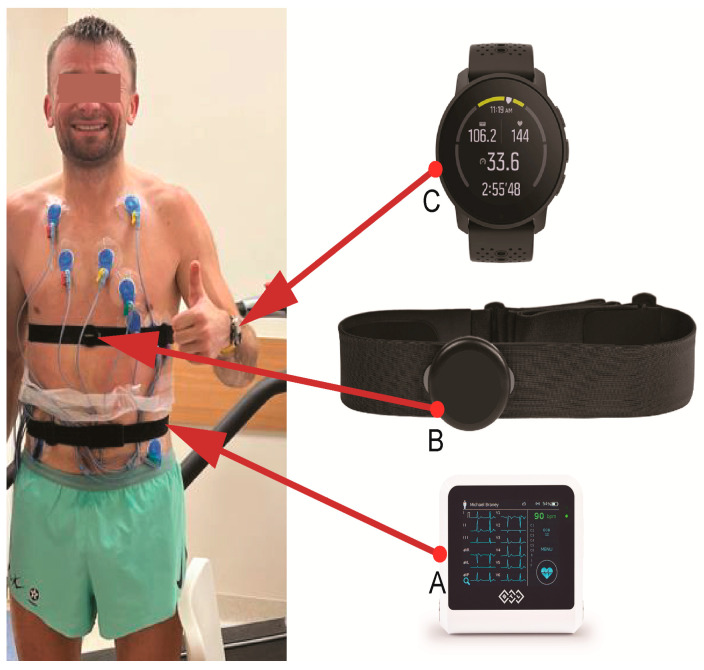
The athlete following an exercise test on a treadmill, wearing the ECG apparatus (BTL Flexi 12 ECG; (**A**) and his heart rate monitor, including the Suunto Smart Heart Rate Belt (**B**) and Smartwatch Suunto 9 Peak (**C**). Only the attachment strap and electrodes of the ECG apparatus are visible because they are attached to the athlete’s back.

**Figure 3 diagnostics-13-02917-f003:**
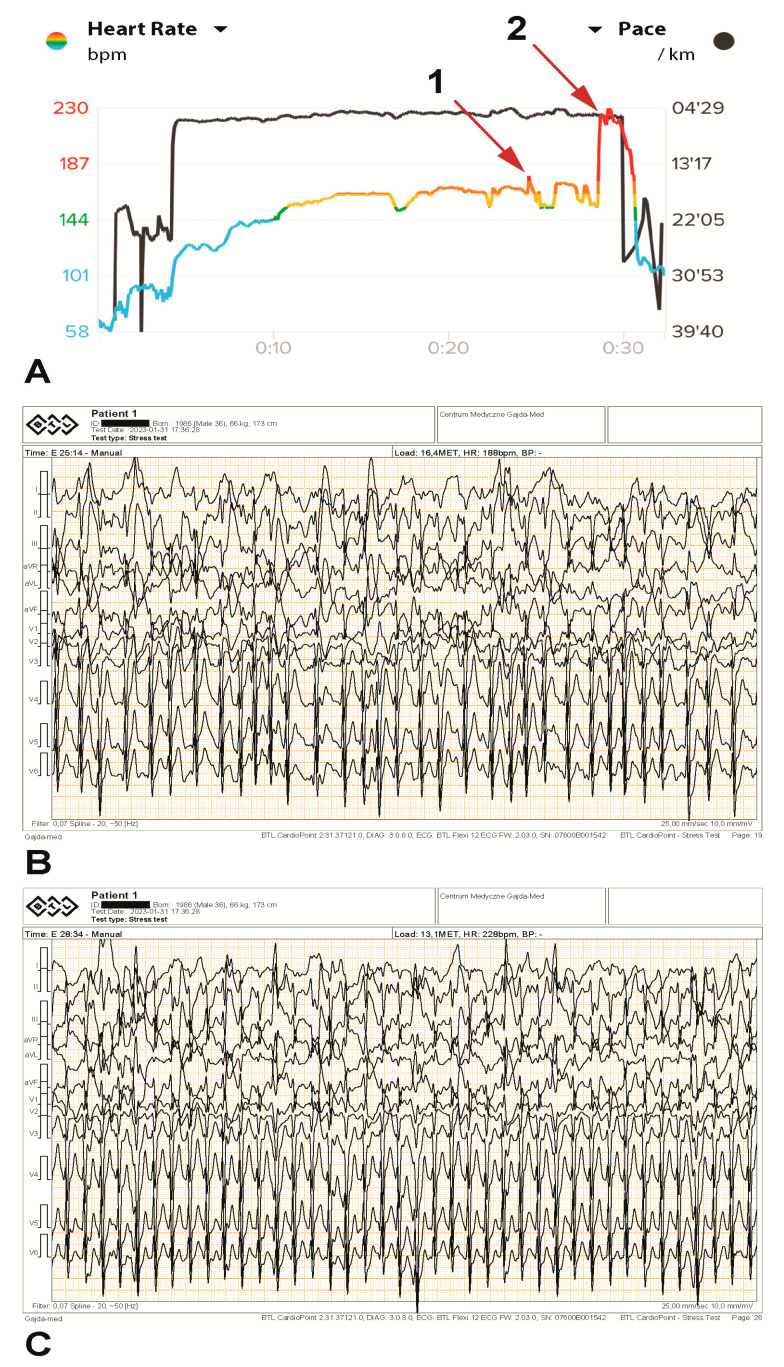
The athlete’s heart rhythm during an exercise test (**A**), including ECG printouts illustrating tachyarrhythmia paroxysms during the test (**B**,**C**). A brief supraventricular tachycardia paroxysm of several seconds was recorded at 187 bpm ((**A**), arrow 1, and (**B**)) and again at 230 bpm ((**A**), arrow 2, and (**C**)) on both devices simultaneously.

**Figure 4 diagnostics-13-02917-f004:**
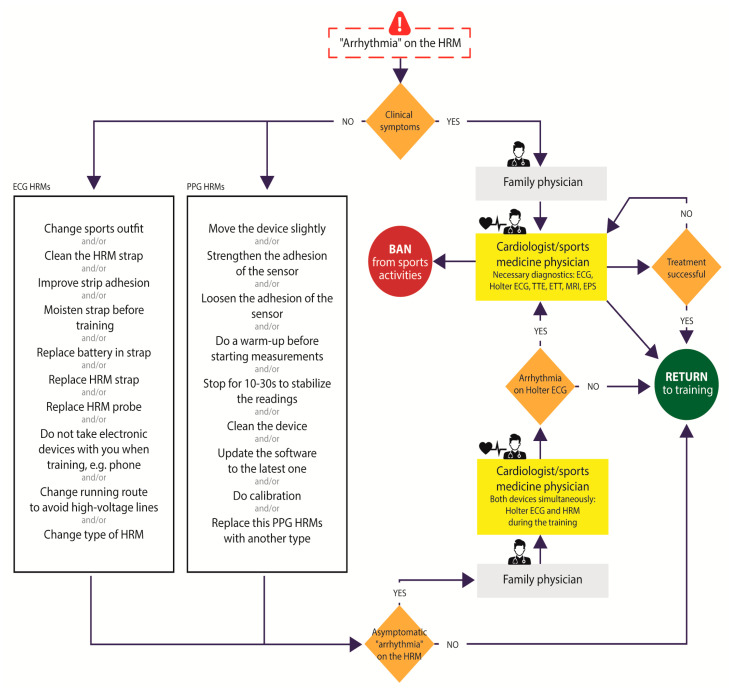
Procedure for managing suspected arrhythmias based on indications by different types of heart rate monitors according to Gajda et al. [26]. Abbreviations: HRM—heart rate monitor, PPG-S—photoplethysmography sensors, ECG-S—electrocardiography sensors, ECG—electrocardiography, TTE—transthoracic echocardiography, ETT—exercise tolerance test, MRI—magnetic resonance imaging, EPS—intracardiac electrophysiology study.

## Data Availability

All data relevant to this study are presented herein or are available in the supplementary materials. Any additional data may be requested from the lead author (Robert Gajda).

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
