# Peer review of "Exercise-Induced Arrhythmia or Munchausen Syndrome in a Marathon Runner?"

_diagnostics, 2023, doi:10.3390/diagnostics13182917_

Round 1
Reviewer 1 Report
I congratulate with authors for this very interesting case report demonistrating the diagnostic difficulties related to exercise-induced arrhythmia and the diagnostic usefulness of sports HR monitor. Exercise induced arrhythmia is a very important issue, however in discussion authors should more discuss about the importance/existence of structural heart disease: in particular. intense exercise-induced remodeling is a potential adaptation of cardiac function and structure and the features of the remodeling may overlap with those of a very early form of arrhythmogenic cardiomyopathy (DOI: 10.1111/jce.14526) or hypertrophyc cardiomyopathy (DOI: 10.1016/j.jacc.2022.07.013 ; DOI: 10.1016/j.ijcard.2021.10.013). Not by chance, at this early stage, it could be difficult to discriminate cardyomyopathy from exercise-induced cardiac adaptation that may develop in normal individuals. This may impact on arrhythmia onset as well, please expand in discussion and cite 3 suggested references
In discussion authors should finally expand how a complete evaluation may be required to identify the pathological features that would also include potential risk of sudden cardiac death during sport or, to avoid the false diagnosis. Please add as well a nice figure
Author Response
"Please see the attachment."

Reviewer 2 Report
accept in present form
Reviewer 3 Report
The format suggests original research rather than a case report and differs from the format indicated in the instruction for authors in Diagnostic. Kindly clarify this.
2. Line 2 (Title): Consider using Munchausen instead of Munchhausen.
3. Line 18 (Abstract): Since palpitation is commonly experienced with physical activity, you may want to clarify to give the connotation of being abnormal. Consider using ‘irregular palpitations’ if that was what the patient reported. Secondly, it may not be necessary to use ‘heart.’
4. Line 78 (Materials and Methods): You may consider using ‘witnessed’ instead of ‘confirm’ since an ECG was not obtained then (line 79).
5. Line 82: You should be specific about the number of stress tests performed. Numerous implies more than two, as indicated in line 87.
6. Line 87: Reconcile ‘two’ with ‘numerous’ (line 82).
7. Line 93: You may want to specifically indicate the ‘additional examinations’ for clarity, as it appears the patient was reasonably examined at the cardiology centers (physical examination, ECG, Echo, CMR, EP study….)
8. Line 171: Although it can be challenging to determine the regularity of the rhythm or otherwise when the ventricular rate is very high, the rhythm depicted in Figure 3C appears to be regular. Kindly clarify. FA paroxysmal should be adequately defined (could mean focal atrial {FA} paroxysmal tachycardia [expected to be regular], or fibrillacion auricular {FA}, an irregular rhythm).
9. Discussing the arrhythmia in the context of bi-atrial enlargement (though, understandably, related to athletes’ hearts) may be relevant. Similarly, a reference to the existing literature on SVT in athletes should be considered, especially when invoking the possibility of paroxysmal AF.
10. Supplemental Figure S1 (ECG): Kindly clarify the criteria for assessing left ventricular hypertrophy, as it doesn’t appear convincing.
11. Supplemental Figure S4 D: The authors may want to comment on the ‘spotty’/’patchy’ LGE in the basal- to mid-lateral wall and inferior septum.
Round 2
Reviewer 1 Report
Manuscript definitely improved! Congratulations⁶